# Deep Neural Network Compression for Plant Disease Recognition

**Ruiqing Wang** [1] **, Wu Zhang** [1,2,*]**, Jiuyang Ding** [1]**, Meng Xia** [1]**, Mengjian Wang** [1]**, Yuan Rao** [1,2]
**and Zhaohui Jiang** [1,2]

[1] School of Information and Computer, Anhui Agricultural University, Hefei 230036, China;
wrq20720820@stu.ahau.edu.cn (R.W.); 20723008@stu.ahau.edu.cn (J.D.); xiameng@ahau.edu.cn (M.X.);
wmj@ahau.edu.cn (M.W.); raoyuan@ahau.edu.cn (Y.R.); jiangzh@ahau.edu.cn (Z.J.)
[2] Anhui Province Key Laboratory of Smart Agricultural Technology and Equipment,
Anhui Agriculture University, Hefei 230036, China
* Correspondence: zhangwu@ahau.edu.cn

**Abstract:** Deep neural networks (DNNs) have become the de facto standard for image recognition tasks, and their applications with respect to plant diseases have also obtained remarkable results. However, the large number of parameters and high computational complexities of these network models make them difficult to deploy on farms in remote areas. In this paper, focusing on the problems of resource constraints and plant diseases, we propose a DNN-based compression method. In order to reduce computational burden, this method uses lightweight fully connected layers to accelerate reasoning, pruning to remove redundant parameters and reduce multiply–accumulate operations, knowledge distillation instead of retraining to restore the lost accuracy, and then quantization to compress the size of the model further. After compressing the mainstream VGGNet and AlexNet models, the compressed versions are applied to the Plant Village dataset of plant disease images, and a performance comparison of the models before and after compression is obtained to verify the proposed method. The results show that the model can be compressed to 0.04 Mb with an accuracy of 97.09%. This experiment also proves the effectiveness of knowledge distillation during the pruning process, and compressed models are more efficient than prevalent lightweight models.

**Keywords:** deep neural networks; plant disease recognition; network pruning; knowledge distillation; model quantization



## 1. Introduction

With the development of computational systems in recent years, especially the rapid progress of graphics processing units (GPUs) [1], deep learning (DL) models [2] have made remarkable achievements in many fields, e.g., natural language processing [3], machine translation [4], medical image analysis [5], and many others. Convolutional neural networks (CNNs), as the basic tools of DL, have been widely applied because of their ability to automatically extract features and process high-dimensional images better than other approaches [6–8]. In addition, the same is true for the application of CNNs in agriculture; they can always achieve outstanding results, whether in fruit counting [9,10], plant phenotyping [11], or other applications as discussed in the surveys [12,13]. Plant disease identification [14] is also a hot topic in the agriculture-based DL since it directly affects the yields of crops and indirectly relates to human economic benefits. However, it is of high complexity to determine whether plant leaves are diseased by optical observation, and even domain experts cannot identify certain diseases with high accuracy, which also requires considerable manpower and time. The great successes of CNNs in image recognition [15]. Also make the identification of plant leaf diseases with these networks preferable and have led to significant breakthroughs.

Regarding the development of CNNs, neural networks have been developed from the original LeNet [16] to current networks with wider, deeper, and more parameters, such

as AlexNet [15] and VGGNet [17]. Although this has yielded greater progress in terms of accuracy, it also brings a large number of parameters, slow reasoning speed, and large memory footprints. Moreover, these problems are amplified in agriculture. Since farms are usually located in resource-constrained areas, the deployment of these models on remote devices would be limited by network bandwidth and delays. On the edge devices, such as mobile devices and Internet of Things devices, due to the characteristics of these resource-constrained devices [18], such massive neural networks cannot be effectively operated. Therefore, it is particularly critical to compress neural networks simply and efficiently.

In the agricultural field, there are few works that focus on the sizes, processing speeds, and resource constraints of neural network models, which are exactly the issues that should be considered. Researchers in [19] deployed a lightweight neural network after knowledge distillation on an agricultural robot platform to distinguish between weeds and crops. In [20–23], authors used lightweight CNNs to identify diseased crop leaves for easier deployment on embedded devices. However, in a related study of model compression [24], a lightweight network was only one part of the solution. Therefore, a simpler, faster, and more efficient neural network can be obtained by combining other related methods while incurring almost no accuracy loss. In this study, pruning, knowledge distillation, and quantization, which are universally applicable methods, are combined to obtain lower computational burdens and fewer parameters by compressing the lightweight VGGNet and AlexNet, which are then applied to the leaf disease images in the Plant Village dataset [25,26].

The rest of this article is organized as follows: Section 2 briefly describes the latest model compression techniques. Section 3 presents the proposed method and process in detail. Then, the experimental dataset and its settings are introduced, and the results of the proposed method are shown in Section 4. Finally, the work of this study is summarized, and thoughts are put forward regarding future work, which may be helpful for other researchers performing related work, in Section 5.

## 2. Related Works

In the field of DL, the pursuit of model accuracy is an important aspect, but the main challenges of deep neural networks (DNNs) also include determining how to reduce the number of model parameters (the size of the model) and the computational cost. Model compression is a software method, and the application cost is low. Compression does not conflict with hardware acceleration, and the two processes can complement each other, so the resulting models can be better deployed on cloud servers or embedded devices. Researchers in [27,28] confirmed that most of the parameters in a given network model are redundant, and it is very possible to establish a "simple" network by removing redundant parameters without affecting the accuracy. The obtained neural network model has lower complexity and can be deployed and applied more conveniently.

### 2.1. Pruning

2.1.1. Pruning Granularity

As the main method of model compression, pruning can be classified into fine-grained and coarse-grained pruning according to the pruning granularity.

Fine-grained pruning, i.e., unstructured pruning, where the pruning granularity is a single neuron, can remove unimportant neurons according to different criteria. In [27,29], Hessian matrices were proposed for the loss function relative to the weights to remove unimportant connections. However, due to the complexity of the calculation, this also brought additional computational costs. In [30], the weights were set to 0 when they were below a certain threshold, and dense matrices were transformed into sparse matrices to accelerate the calculation. However, it is difficult to achieve substantive acceleration without hardware support and specialized software libraries [31]. Therefore, most of the existing pruning studies have focused on structured pruning, which can also achieve an unstructured compression ratio and acceleration in some cases.

Coarse-grained pruning, i.e., structured pruning, removes an entire structure, and it can accelerate and compress models without special hardware. Among the aspects of structured pruning, since multiply–accumulate (MAC) operations are mostly concentrated in filters, pruning at the filter level is an important consideration. The removal of insignificant filters can reduce the memory requirements and computational budgets of the model. Filter-level pruning can be expressed as an optimization problem:

$$
\min_{\delta_{ij}} \sum_{i=1}^{K} \sum_{j=1}^{n_i} \delta_{ij} \mathcal{L}\left(\mathbf{w}_j^i\right),
$$

$$
\text{s.t.} \sum_{j=1}^{n_i} \delta_{ij} = n_{i2},
$$

(1)

where $\mathbf{w}_j^i$ represents the $j$th filter in the $i$th convolution layer, and $\delta_{ij}$ is an indicator. When $\mathbf{w}_j^i$ is grouped in the removed filter set, its value is 1, and when it is in the reserved filter set, its value is 0. $\mathcal{L}(\cdot)$ is used to estimate the importance of the filters. Minimizing Equation (1) is equivalent to removing the least important filters in $n_{i2}$. Therefore, determining how to measure the importance of filters, that is, how to design $\mathcal{L}(\cdot)$ becomes a top priority.

In [32], the authors sorted the filters of each convolution layer according to the L1-norm and considered that the smaller the L1-norm of a given layer was, the lower its importance was, which meant that it should be removed. The authors in [33] proposed ThiNet, and the statistical information of the next layer of feature maps was applied to the pruning of the current convolution layer. The authors in [34] proposed applying L1 regularization to the scaling factors of batch normalization (BN) layers to remove filters with lower scaling factors. In [35], the authors considered that the ranks of the feature maps generated by convolution layers determined the amount of information contained, i.e., whether the filters were important, and obtained state-of-the-art results.

### 2.1.2. Pruning Strategies

Pruning strategies can be classified as one-shot and iterative pruning.

One-shot pruning involves achieving the preset sparsity with just one step and then retraining the model. The process is shown in Figure 1.



**Figure 1.** The network model performs pretraining first and then prunes once according to the preset evaluation criteria to achieve the target pruning rate. Finally, the model is fine-tuned to obtain the final model.

Although this strategy is very simple and does not bring additional hyperparameters, it can easily cause accuracy losses that cannot be repaired. In contrast, iterative pruning prunes only a portion of the filters each iteration and retrains the model several times; then, pruning and retraining cycles are repeated to achieve the target sparsity. The authors compared the accuracy of the models obtained after iterative pruning and one-shot pruning in [32], and the results showed that iterative pruning is more effective in compensating for the loss incurred after removing the filter. Researchers in [36] proposed the use of step sparsity to determine the number of filters to be removed in each iteration of pruning.

### 2.2. Knowledge Distillation

Knowledge distillation, also known as teacher–student training, was proposed in [37]. An experiment demonstrated that a well-trained network uses its output soft labels to guide a simple, small student network through training, and dark knowledge can be easily transferred to the student network without changing its structure. In [38], authors proposed an ensemble of teacher networks to improve the generalization ability of a student network.

In [39], the authors used the attention maps of the middle layers of a teacher network to guide the training of the student network, aiming at enabling the student network to learn the feature maps of the teacher more effectively. Researchers in [40] found that if the given teacher and student networks are similar in structure, the student network more easily learns the knowledge of the teacher network, thus saving time when training a complex model.

### 2.3. Quantization

One method of quantization is to cluster the model weights, which are classified into the same category and can share weight values, to realize the compression of the model; this approach is called vector quantization. In [41], the authors used K-means clustering to implement vector quantization, as this method can achieve a 16–24 time compression ratio, and the accuracy loss lies in an acceptable range. However, the storage of the shared weight codebook and its computation also requires additional resources.

Another method of quantization is to approximate the weights of 32-bit floating-point numbers with fewer bits (such as 16-bit or 8-bit weights), which is called fixed-point quantization. Due to the reduction in the representation bits of weights, the size of the network model can be reduced. Researchers in [42] showed that the use of uint-8 representations to represent the original bits could achieve effective acceleration without sacrificing accuracy. The use of one-bit data to replace the original weight bits is called the binary neural network [43]; although such a network can greatly reduce the network computing and storage costs, determining how to maintain the accuracy after quantization becomes a great challenge.

The application of quantization in the inference stage can accelerate forward propagation and further improve the compression rate and can be combined with other compression methods. For example, in [44], the authors proposed a compression method combining pruning, quantization, and Huffman encoding, and the model could be compressed up to 49 times.

### 2.4. Lightweight Networks

Unlike pruning and quantization, lightweight networks directly design an efficient model to solve the given problem.

MobileNet [45] decomposes the standard convolution operation into depthwise convolution and pointwise convolution. Depthwise convolution is a convolution operation performed in each independent channel, and pointwise convolution is a $1 \times 1$ convolution operation, both of which can greatly reduce the number of required calculations without affecting model accuracy. SqueezeNet [46] achieves approximately equal accuracy to that of AlexNet on ImageNet; moreover, the number of parameters is 50 times less than that of AlexNet.

### 2.5. DL Architectures for Plant Disease Detection

Using new or modified DL architectures has become a mainstream method for plant disease detection. In [47], the authors proposed a lightweight model and strengthened the generalization performance of the model on a plant disease dataset by transfer learning, which can achieve 89.70% accuracy. In [48–50], authors used different DL architectures (such as AlexNet, VGGNet, GoogleNet, and ResNet) to detect plant disease, and achieved promising results, which also promoted the application of DL in agriculture. In [51], the authors introduced a novel DL architecture considering the spot attention mechanism of apple leaf disease and proved its performance is better than traditional DL models.

In this paper, unlike the traditional research that only uses lightweight models [20–23,47], we combine pruning, distillation, and quantization methods to minimize the size of the model while ensuring accuracy. Compared with existing lightweight models, our proposed model compression method on VGGNet and AlexNet yields more competitive results (see Section 4.3 for details).

### 3. Methodology

This paper proposes a model compression method for plant disease identification, which includes lightweight, iterative pruning, knowledge distillation, and quantization. The specific flowchart is shown in Figure 2.

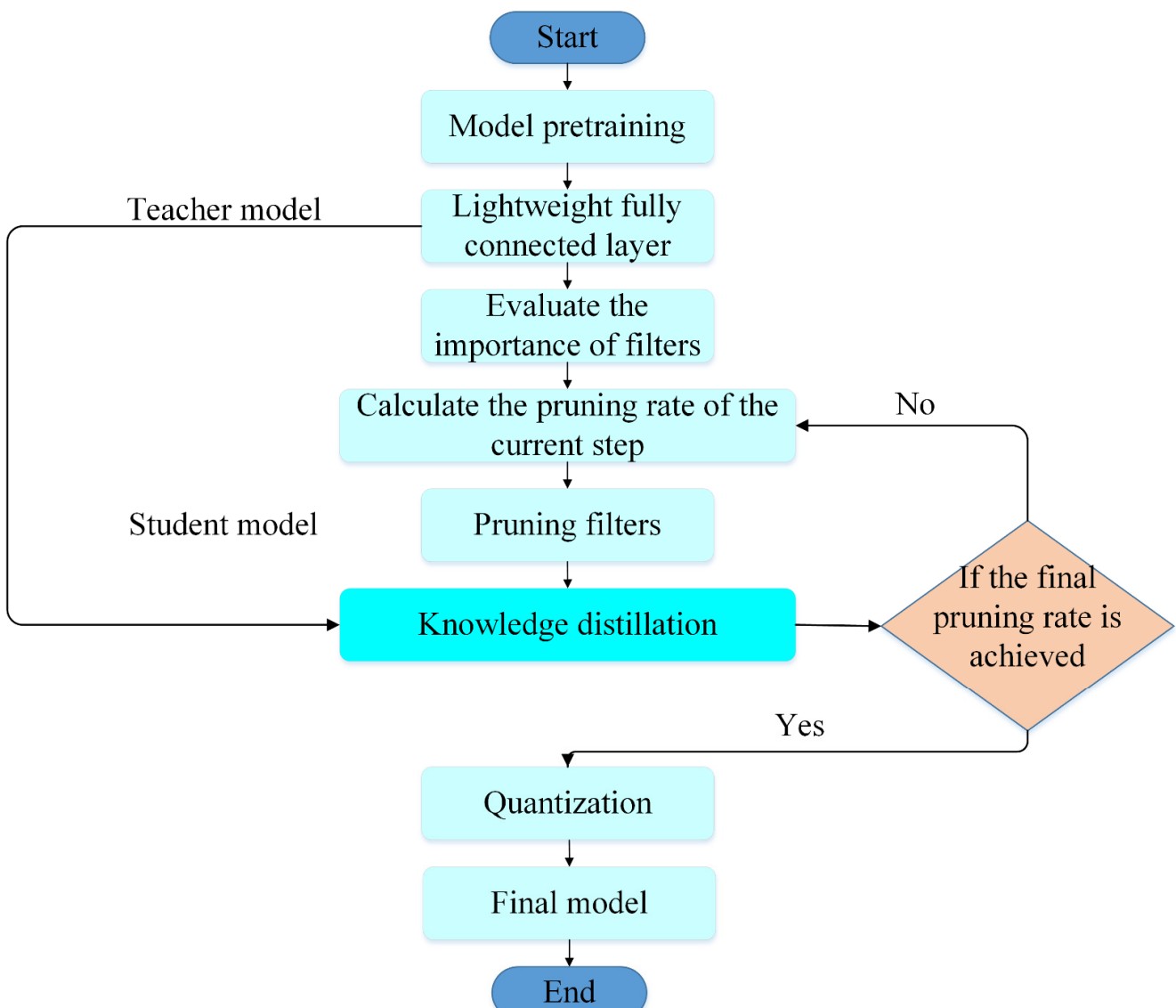

**Figure 2.** The flowchart of the proposed compression method.

### 3.1. Lightweight on Fully Connected Layers

Before using correlation compression methods, the weight of the network model should be reduced first. In CNNs, the fully connected layers incur a large number of parameters. As shown in Figure 3, the fully connected layers of VGGNet and AlexNet account for 89.36% and 95.96% of the total number of parameters, respectively. Therefore, to simplify the models, it is necessary to compress the fully connected layers. However, pruning the fully connected layers results in weight sparsity. If there is no special hardware or corresponding software libraries, the acceleration effect is inconspicuous. To solve this problem, in this study, global average pooling (GAP) [52] is used to reduce the burden of the model, as shown in Figure 4. The feature maps of the last convolution layer are pooled to obtain the results so that the extracted features and classification output are directly related. On the one hand, reducing the large number of parameters can reduce the

computational cost; on the other hand, it can prevent overfitting, and there is little effect on accuracy.

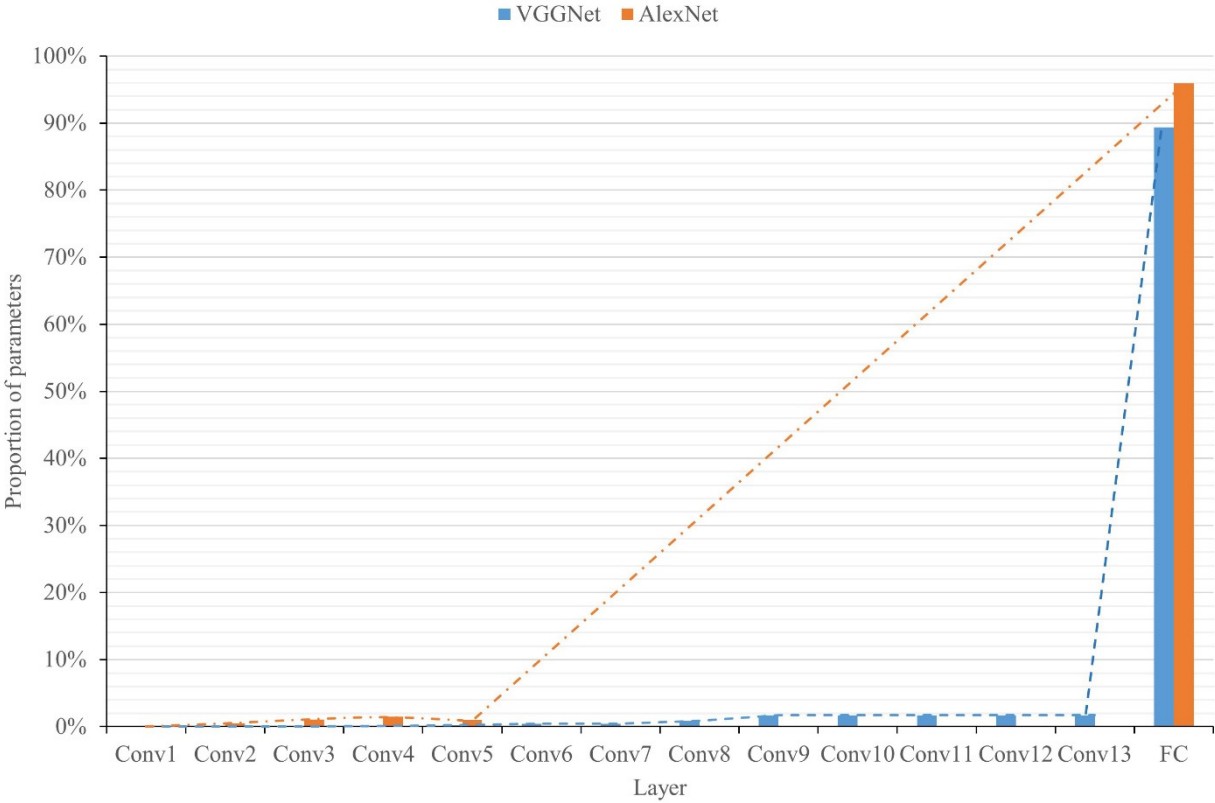

**Figure 3.** The proportions of parameters in each layer of VGGNet and AlexNet. VGGNet has 13 convolution layers, AlexNet has 5 convolution layers, and the FC refers to all the parameters of the fully connected layers.

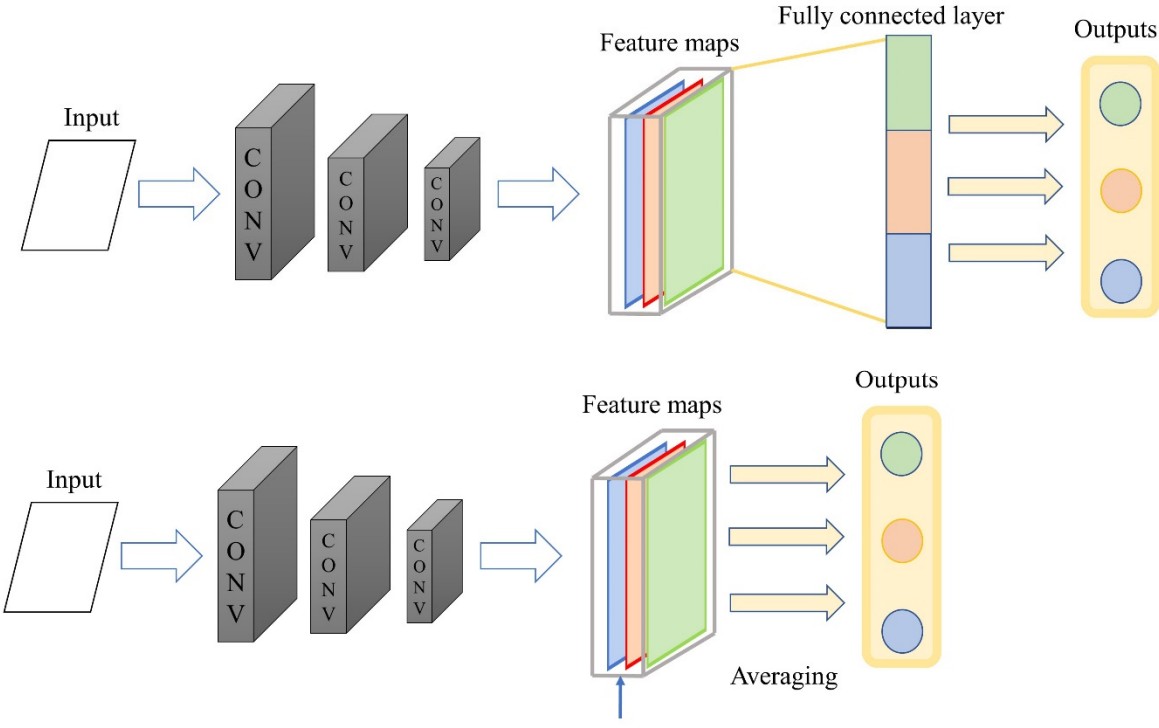

**Figure 4.** Replacing fully connected layers with GAP.

### 3.2. Iterative Pruning with Knowledge Distillation

In this study, filter-level structured pruning is applied, where the standard for measuring the importance of the filters is the ranks of the model feature maps [35]; additionally, knowledge distillation is used instead of retraining. The core flow of the proposed algorithm is shown in Algorithm 1.

### 3.2.1. Iterative Pruning

During the process of pruning, a small batch of input images is used to accurately predict the expected rank of the examined feature map, and Equation (1) can be more specifically expressed as:

$$\min_{\delta_{ij}} \sum_{i=1}^{K} \sum_{j=1}^{n_i} \delta_{ij} \mathcal{L}\left(\mathbf{w}_j^i\right) \sum_{K=1}^{G} \mathbf{Rank}\left(\mathbf{o}_j^i(K,:,:)\right),$$

$$\text{s.t.} \sum_{j=1}^{n_i} \delta_{ij} = n_{i2},$$

(2)

where $\mathbf{o}_j^i(K,:,:)$ is the matrix of the feature map of input image $K$ generated by $\mathbf{w}_j^i$. $\mathbf{Rank}(\cdot)$ is the rank of the feature map matrix that evaluates the average rank across $G$ images.

To distinguish from one-shot pruning, a certain number of filters are removed at each step during iterative pruning. Therefore, in this experiment, a polynomial function Equation (3) is proposed to obtain the step pruning rate.

$$S_t = 1 - \left(1 - S_f\right)^{\frac{t+2}{t_f+2}}$$

(3)

where $S_t$ and $S_f$ are the current and target pruning rates, respectively. $t$ is the current pruning step, and $t_f$ is the total number of training steps. In this way, one-shot pruning is avoided, and the pruning of the filters can be completed within the limited number of training steps, which realizes a tradeoff between training time and accuracy.

---

**Algorithm 1.** Framework of iterative pruning with knowledge distillation

---

$/*$ **Initialization** $*/$
1: Convolution layer index to prune: $L$;
2: Student model pruning rate at step $t$: $S_t\%$;
3: Student model final pruning rate at step $t_f$: $S_f\%$;
$/*$ **Pretraining** $*/$
4: Training lightweight model $T$;
5: Pretrained student model $S$: $S \leftarrow T$;
$/*$ **Iterative pruning** $*/$
6: $S_t = 1 - \left(1 - S_f\right)^{\frac{t+2}{t_f+2}}$
7: Pruning $(S_t - S_{t-1})\%$ filters of $L$ with feature map ranks;
$/*$ **Knowledge distillation** $*/$
8: Retraining model $S$ with knowledge distillation with $T$;
9: If $(S_t < S_f)$ then Goto Iterative pruning
$/*$ **Final model** $*/$
10: Return student model $S$;

---

### 3.2.2. Retraining

In the traditional three-stage iterative pruning process, after each pruning step, the model is retrained to restore the accuracy loss incurred by pruning. In this paper, knowledge distillation [37] is used instead of retraining to obtain a better precision recovery effect, as shown in Figure 5.

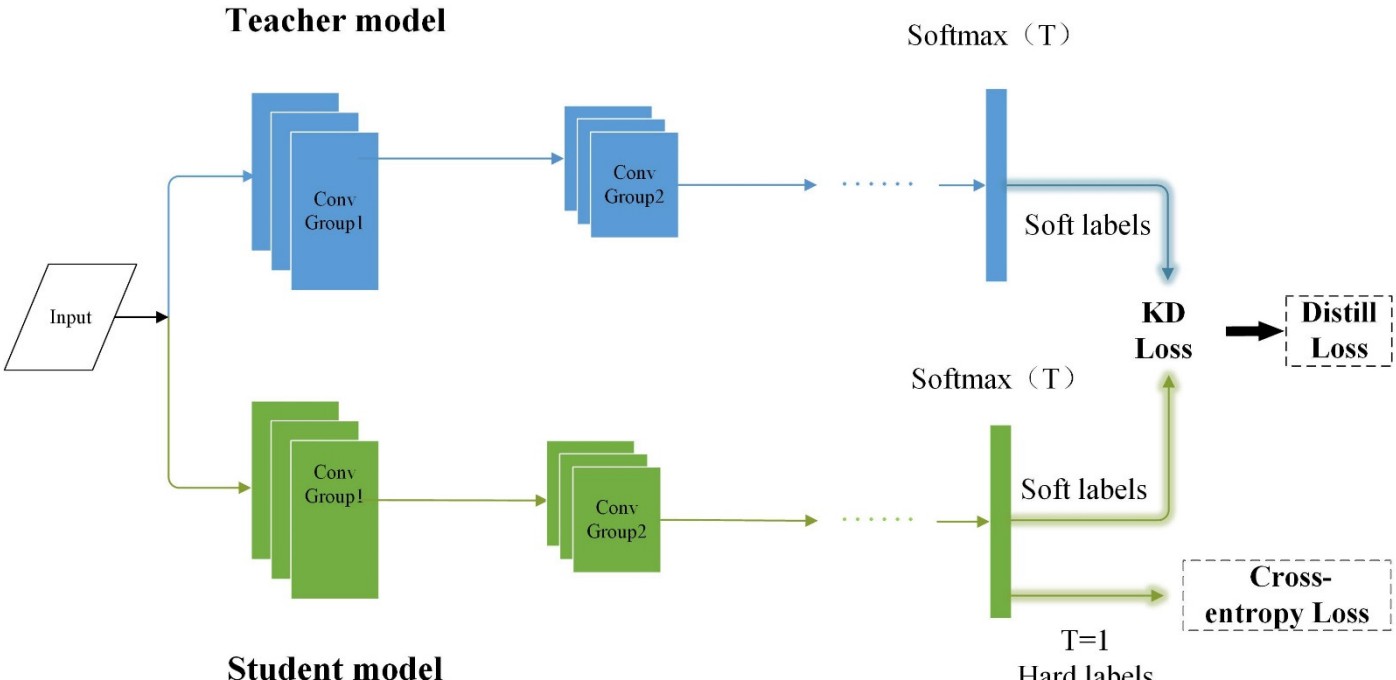

**Figure 5.** Using the outputs of the teacher network to guide the outputs of the student network.

In knowledge distillation, the teacher model uses output soft labels to enable the student model to learn dark knowledge; this process can be defined as:

$$L_{kd} = 2T^2 \times L_{KL}(S_{Stu}, S_{Tea}) \tag{4}$$

where the temperature parameter $T$ is used to control the smoothness of the output to preferably transfer the knowledge of the teacher network. $L_{KL}$ is the Kullback–Liebler (KL) divergence between the soft labels of the two network models. The soft label output $S_{stu}$ of the student network can be given by:

$$S_{Stu} = \frac{\exp(z_s/T)}{\sum\limits_{i=1}^{n} \exp\left(z_S^{(i)}/T\right)} \tag{5}$$

where $z_S$ is the output of the student network without a softmax layer. Equation (5) controls the degree of smoothness of the output, and it is the softmax function when $T = 1$. $S_{Tea}$ can also be calculated in this way.

Under the effect of knowledge distillation, the total loss function of the student network model becomes:

$$L_{\text{student}} = \alpha L_{kd} + \beta \text{CrossEntropy}(z_S, y_{Ture}) \tag{6}$$

where $y_{True}$ denotes the ground-truth label of the output, and the latter term is the classic cross-entropy loss. $\alpha$ and $\beta$ are weight hyperparameters.

*3.3. Fixed-Point Quantization*

Quantization is applied after training, and uint-8 bit-based fixed-point quantization without data calibration is used to compress the matrix of weights.

## 4. Experimental Evaluation

*4.1. Experimental Settings*

4.1.1. Datasets

This paper used 54,306 images of diseased and healthy plants under controlled conditions from a public database [25,26], which were obtained by researchers using a standard digital camera with automatic mode. This database includes 38 different classes, each of which contains disease or health data corresponding to a certain plant. Table 1 lists the information of these 38 classes, including 14 plants and 26 diseases (some plants have only healthy images). Eighty percent of the images were randomly selected as the training set, while the remaining 20% were selected as the test set.

**Table 1.** Related information about the database images.

| Class | Plant Common Name | Plant Scientific Name | Disease Common Name | Disease Scientific Name | Images (Number) |
|---|---|---|---|---|---|
| C_1 | Apple | Malus domestica | – | – | 1645 |
| C_2 | Apple | Malus domestica | Apple scab | Venturia inaequalis | 630 |
| C_3 | Apple | Malus domestica | Black rot | Botryosphaeria obtusa | 621 |
| C_4 | Apple | Malus domestica | Cedar apple rust | Gymnosporangium juniperi-virginianae | 275 |
| C_5 | Blueberry | Vaccinium spp. | – | – | 1502 |
| C_6 | Cherry (and sour) | Prunus spp. | – | – | 854 |
| C_7 | Cherry (and sour) | Prunus spp. | Powdery mildew | Podosphaera spp. | 1052 |
| C_8 | Corn (maize) | Zea mays | – | – | 1162 |
| C_9 | Corn (maize) | Zea mays | Cercospora leaf spot | Cercospora zeae-maydis | 513 |
| C_10 | Corn (maize) | Zea mays | Common rust | Puccinia sorghi | 1192 |
| C_11 | Corn (maize) | Zea mays | Northern leaf blight | Exserohilum turcicum | 987 |
| C_12 | Grape | Vitis vinifera | – | – | 423 |
| C_13 | Grape | Vitis vinifera | Black rot | Guignardia bidwellii | 1180 |
| C_14 | Grape | Vitis vinifera | Esca (Black measles) | Phaeomoniella chlamydospora | 1383 |
| C_15 | Grape | Vitis vinifera | Leaf blight | Pseudocercospora vitis | 1076 |
| C_16 | Orange | Citrus sinensis | Huanglongbing | Candidatus Liberibacter | 5507 |
| C_17 | Peach | Prunus persica | – | – | 360 |
| C_18 | Peach | Prunus persica | Bacterial spot | Xanthomonas campestris | 2297 |
| C_19 | Pepper, bell | Capsicum annuum | – | – | 1477 |
| C_20 | Pepper, bell | Capsicum annuum | Bacterial spot | Xanthomonas campestris | 997 |
| C_21 | Potato | Solanum tuberosum | – | – | 152 |
| C_22 | Potato | Solanum tuberosum | Early blight | Alternaria solani | 1000 |
| C_23 | Potato | Solanum tuberosum | Late blight | Phytophthora infestans | 1000 |
| C_24 | Raspberry | Rubus spp. | – | – | 371 |
| C_25 | Soybean | Glycine max | – | – | 5090 |
| C_26 | Squash | Cucurbita spp. | Powdery mildew | Erysiphe cichoracearum, Sphaerotheca fuliginea | 1835 |
| C_27 | Strawberry | Fragaria spp. | – | – | 456 |
| C_28 | Strawberry | Fragaria spp. | Leaf scorch | Diplocarpon earlianum | 1109 |
| C_29 | Tomato | Lycopersicum esculentum | – | – | 1591 |
| C_30 | Tomato | Lycopersicum esculentum | Bacterial spot | Xanthomonas campestris pv. Vesicatoria | 2127 |
| C_31 | Tomato | Lycopersicum esculentum | Early blight | Alternaria solani | 1000 |
| C_32 | Tomato | Lycopersicum esculentum | Late blight | Phytophthora infestans | 1909 |
| C_33 | Tomato | Lycopersicum esculentum | Leaf mold | Fulvia fulva | 952 |
| C_34 | Tomato | Lycopersicum esculentum | Septoria leaf spot | Septoria lycopersici | 1771 |
| C_35 | Tomato | Lycopersicum esculentum | Spider mites | Tetranychus urticae | 1676 |
| C_36 | Tomato | Lycopersicum esculentum | Target spot | Corynespora cassiicola | 1404 |
| C_37 | Tomato | Lycopersicum esculentum | Tomato mosaic virus | Tomato mosaic virus (ToMV) | 373 |
| C_38 | Tomato | Lycopersicum esculentum | TYLCV | Begomovirus (Fam. Geminiviridae) | 5357 |
| *TOTAL:* | | | | | *54,306* |

In order to reduce overfitting and improve the generalization ability of the model, rotation mirror and mirror symmetry were used for the training set, as shown in Figure 6. Eighty percent of the images were randomly selected as the training set, while the remaining 20% were selected as the test set.

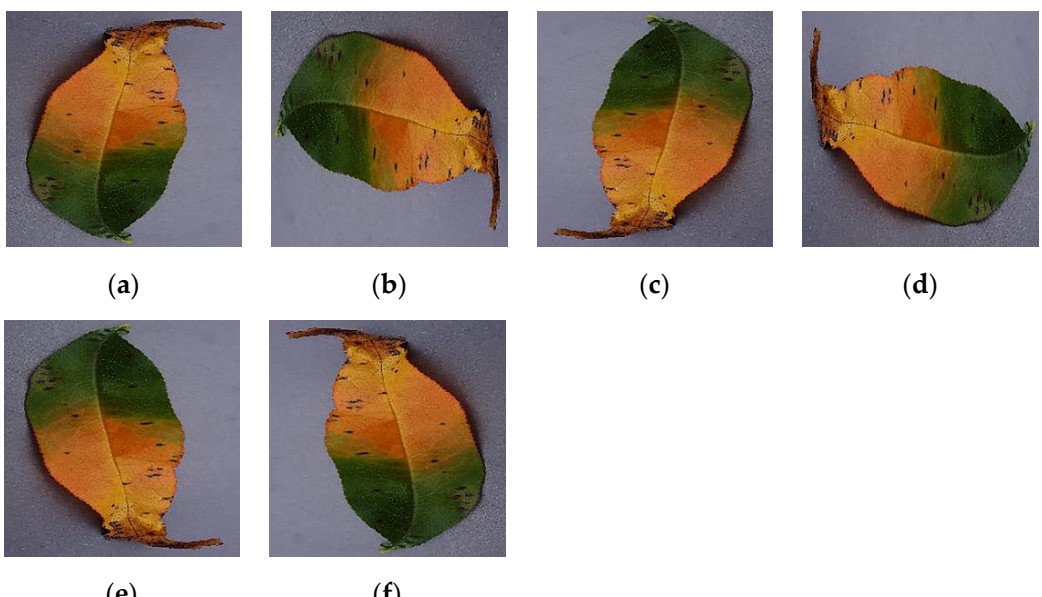

**Figure 6.** Examples of rotation mirror and mirror symmetry used for Peach Bacterial Spot image: (**a**) initial; (**b**) rotated 90°; (**c**) rotated 180°; (**d**) rotated 270°; (**e**) vertical mirror symmetry; (**f**) horizontal mirror symmetry.

### 4.1.2. Configurations

In this experiment, the popular AlexNet and VGGNet-16 models with BN layers were selected for lightweight, pruning, distillation, and quantization and applied to the public database of plant diseases.

During the training process, the initial learning rate is 0.01, and the learning rate drops by 50% every 20 epochs. The stochastic gradient descent with weight-decay = $5 \times 10^{-4}$ and momentum = 0.9 is selected as the optimizer. The total number of epochs used to train the original model is 60, and the batch size is set to 32.

For Equation (2), $G = 640$ (i.e., 20 batches) pictures are used to estimate the average rank of the feature map generated by each filter. Then, during the iterative pruning process of Equation (3), the final pruning rate $S_f$ is set to 0.85, 0.9, and 0.95, and $t_f$ is set to 100. When $t = 0$, pruning begins, and when $t = 100$, the final pruning rate is obtained. After pruning all convolution layers, training is continued for 40 additional epochs to restore accuracy, and the learning rate is also decreased by 50% when training is half complete. The detailed process of sparsity-level variation across the training steps during pruning is shown in Figure 7. To prevent the introduction of excessive hyperparameters, the final pruning rate of each layer is the same, and due to the introduction of GAP, pruning does not operate on the last convolution layer. In addition, during knowledge distillation, for Equations (4) and (6), $T = 10$, $\alpha = 0.1$, and $\beta = 0.9$. All the experiments were carried out on PyTorch 1.6, CUDA 10.1, and CUDNN 7.6.5 with an NVIDIA GeForce GTX 1080Ti GPU and an Intel i5 10,600 k CPU.

### 4.2. Experimental Results

In this paper, the numbers of parameters and floating-point operations (FLOPs) are used as the criteria for measuring the model size and computing requirements, which represent the memory occupation and the number of additions and multiplications required for forward propagation; these are common criteria in most related studies. Since the size of each input image is closely related to the required FLOPs, it is necessary to change $224 \times 224 \times 3$ to $56 \times 56 \times 3$. As shown in Table 2, within the acceptable range of accuracy decline, the model is accelerated by approximately 15 times.

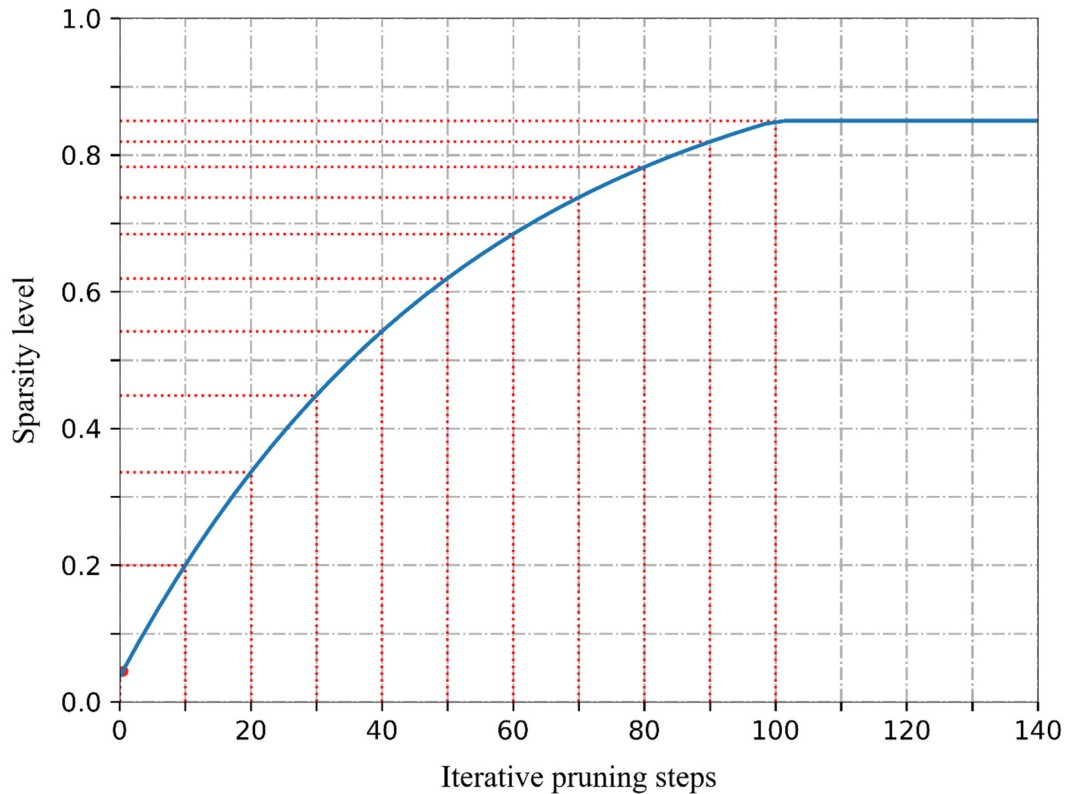

**Figure 7.** The sparsity changes with the increase in the number of pruning steps. A final sparsity of 0.85 is obtained at step 140. The dotted line represents the pruning step and the corresponding sparsity level.

**Table 2.** The changes in the accuracy and FLOPs of VGGNet after changing the sizes of the input pictures.

| VGGNet | Input Size | | |
|---|---|---|---|
| | $224 \times 224 \times 3$ | $112 \times 112 \times 3$ | $56 \times 56 \times 3$ |
| FLOPS | 15.53 G | 3.97 G | 1.06 G |
| Accuracy | 99.59% (+0.0%) | 99.45% (−0.14%) | 99.43% (−0.16%) |

Table 3 provides various pieces of information about two DL models with and without GAP. This shows that the decrease in model accuracy is within the acceptable range when using GAP, and the number of parameters is greatly reduced. The accuracies of VGGNet and AlexNet decrease by only 0.06% and 0.03%, respectively, reaching 99.37% and 98.15%.

**Table 3.** Comparison between the accuracy and parameters of VGGNet and AlexNet with and without GAP.

| Model | VGGNet | VGGNet (GAP) | AlexNet | AlexNet (GAP) |
|---|---|---|---|---|
| Size | 512.79 Mb | 47.83 Mb | 218.06 Mb | 7.51 Mb |
| Accuracy | 99.43% (+0.0%) | 99.37% (−0.06%) | 99.18% (+0.0%) | 98.15% (−0.03%) |

4.2.1. Filter Pruning

Table 4 presents the accuracies, model sizes, and FLOPs of the two architectures at different pruning rates. In addition, VGGNet has better adaptability than AlexNet. Due to the complicated network architecture of VGGNet, even at a 95% pruning rate, the model



accuracy is reduced by 2.17%, and the model can be compressed by a factor of 299 and accelerated by a factor of 284. At a pruning rate of 90%, the accuracy is 98.65%, which is a decrease of 0.72%. While its accuracy is almost unchanged at a pruning rate of 85%, the number of parameters is reduced by a factor of 41, and the model is accelerated by a factor of 40.

**Table 4.** Change in the number of parameters, accuracy, and FLOPs of VGGNet and AlexNet under different pruning rates.

| Model | | Original | 85% | 90% | 95% |
|---|---|---|---|---|---|
| | | | Pruning | | |
| VGGNet (GAP) | Size | 47.83 Mb | 1.18 Mb | 0.56 Mb | 0.16 Mb |
| | Accuracy | 99.37% (+0.0%) | 99.05% (−0.32%) | 98.65% (−0.72%) | 97.20% (−2.17%) |
| | FLOPs | 920,230,062 | 22,974,968 | 10,764,786 | 3,240,726 |
| AlexNet (GAP) | Size | 7.51 Mb | 0.23 Mb | 0.12 Mb | 0.04 Mb |
| | Accuracy | 98.15% (+0.00%) | 94.63% (−3.52%) | 92.19% (−5.96%) | 87.32% (−10.83%) |
| | FLOPs | 21,628,408 | 1,085,788 | 666,896 | 322,200 |

When the pruning rates of AlexNet are 85%, 90%, and 95%, its accuracy drops from 98.15% by 3.52%, 5.96%, and 10.83%, respectively, while it is compressed by approximate factors of 25, 63, and 188 and accelerated by factors of 20, 32 and 62. Since AlexNet has only five convolution layers, the shallow network architecture causes the filters to have high sensitivity and limited performance, resulting in obvious accuracy losses at a high pruning rate; however, the model size is only 0.04 Mb.

In this experiment, a lightweight network was used as the teacher model before pruning, and a network was used as the student model to perform pruning to further improve the accuracy. Table 5 shows the accuracy comparisons before and after knowledge distillation, which are sufficient to prove the effectiveness of using knowledge distillation. It is also observed that when the pruning rate is 90%, the effect of knowledge distillation is best, and the accuracy can be improved by 0.09% and 0.63% for VGGNet and AlexNet, respectively. When the pruning rate is 95%, the accuracy can be improved by 0.06% and 0.31% for VGGNet and AlexNet, respectively. When the pruning rate is 85%, the accuracy can be increased by only 0.01% and 0.19%.

**Table 5.** Effect of knowledge distillation on the accuracy of VGGNet and AlexNet under different pruning rates.

| Model | | Original | 85% | 90% | 95% |
|---|---|---|---|---|---|
| | | | Pruning | | |
| VGGNet (GAP) | Accuracy | 99.37% | 99.05% | 98.65% | 97.20% |
| | | | Knowledge distillation | | |
| | Accuracy (KD) | – | 99.06% (+0.01%) | 98.74% (+0.09%) | 97.26% (+0.06%) |
| AlexNet (GAP) | Accuracy | 98.15% | 94.63% | 92.19% | 87.32% |
| | | | Knowledge distillation | | |
| | Accuracy (KD) | – | 94.82% (+0.19%) | 92.82% (+0.63%) | 87.63% (+0.31%) |

### 4.2.2. Quantization

In the final step of the proposed method, uint-8 fixed-point quantization is used to further compress the model after pruning and distillation to obtain a higher compression ratio. Table 6 (top) shows that the accuracy of VGGNet is not obviously decreased after quantization and increases by 0.01% at a sparsity level of 90%. When the total parameter

size is only 0.04 Mb, which is reduced by a factor of 1196, the accuracy can still reach 97.09%, with a drop of merely 2.28% from the original accuracy rate. Especially at a sparsity level of 85%, the accuracy of the model can still reach 99.06%.

**Table 6.** Change in the model size and accuracy of VGGNet and AlexNet after quantization.

| Model | | Original | 85% | 90% | 95% |
|---|---|---|---|---|---|
| | | | Pruning | | |
| VGGNet (GAP) | Size | 47.83 Mb | 1.18 Mb | 0.56 Mb | 0.16 Mb |
| | Accuracy | 99.37% (+0.0%) | 99.05% (−0.32%) | 98.65% (−0.72%) | 97.20% (−2.17%) |
| | | **Fixed-point quantization with uint-8 bits.** | | | |
| | Size | _ | 0.29 Mb | 0.14 Mb | 0.04 Mb |
| | Accuracy | _ | 99.06% (−0.31%) | 98.75% (−0.62%) | 97.09% (−2.28%) |
| AlexNet (GAP) | Size | 7.51 Mb | 0.23 Mb | 0.12 Mb | 0.04 Mb |
| | Accuracy | 98.15% (+0.00%) | 94.63% (−3.52%) | 92.19% (−5.96%) | 87.32% (−10.83%) |
| | | **Fixed-point quantization with uint-8 bits.** | | | |
| | Size | _ | 0.06 Mb | 0.03 Mb | 0.01 Mb |
| | Accuracy | _ | 94.75% (−3.40%) | 92.78% (−5.37%) | 87.51% (−10.64%) |

AlexNet also exhibits no significant decrease in accuracy after quantization. In Table 6 (bottom), it can be seen that at a sparsity level of 85%, the model size is reduced by a factor of 125. It can be compressed by a factor of 751 after quantization at a high sparsity level of 95%, and the model size is only 0.01 Mb, but the accuracy is 87.51%, 10.64% lower than the original accuracy rate.

Examining the overall effect of quantization, it can be seen that the accuracy losses of the two kinds of network models after quantization are within the acceptable range, and the model sizes can also be compressed approximately 4 times more. Furthermore, due to the decrease in bits, the MAC operations of the convolution layers become simplified, and the forward propagation speed becomes faster.

### 4.3. Comparsion with Lightweight Model

In order to verify the effectiveness of the proposed method on VGGNet and AlexNet models, we compare compressed versions with the prevalent lightweight model on the same dataset. Table 7 provides various pieces of information about MobileNet in the same experimental environment as VGGNet and AlexNet. By comparing it with Table 3, we can observe that lightweight networks obtain better results than uncompressed networks. However, as shown in Figure 8, more competitive results can be obtained when these networks are compressed and knowledge-distilled. The accuracy of VGGNet is 0.21% higher than that of MobileNetV2 at an 85% pruning rate, and the number of parameters is greatly reduced. From the perspective of model size and FLOPs, AlexNet is more efficient after pruning, although its performance decreases obviously.

**Table 7.** Various pieces of information about MobileNetV1 and MobileNetV2 on the plant disease dataset.

| Model | MobileNetV1 | MobileNetV2 |
|---|---|---|
| Size | 12.38 Mb | 8.67 Mb |
| Accuracy | 99.20% | 98.85% |
| FLOPs | 150,466,304 | 23,550,528 |

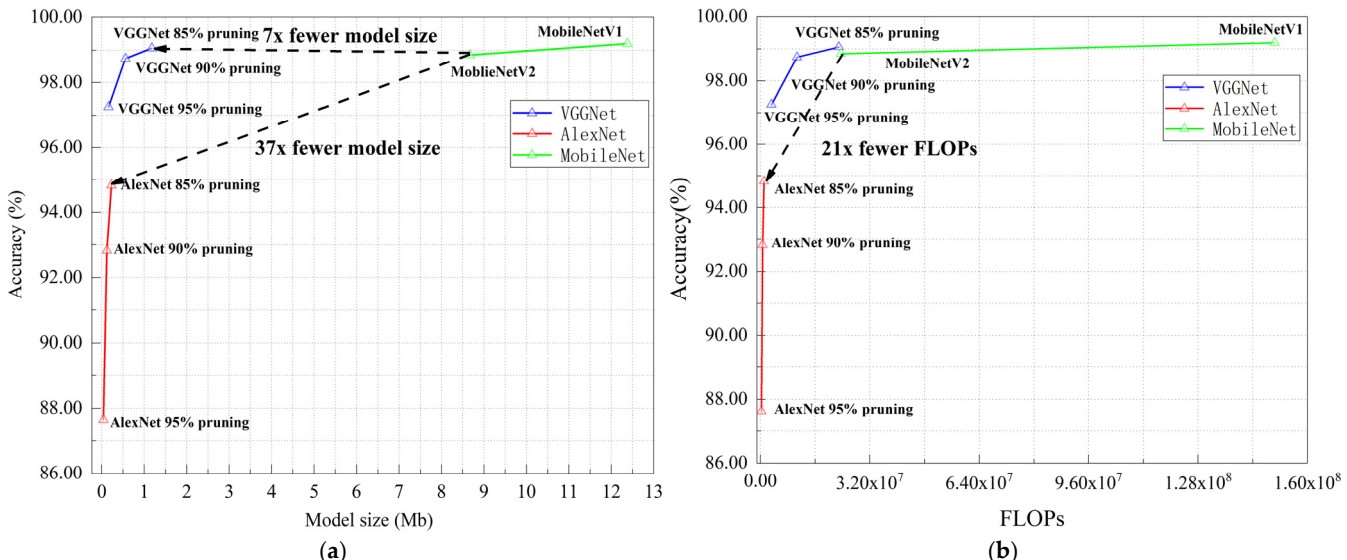

**Figure 8.** Comparison of MobileNet, VGGNet, and AlexNet accuracy on the plant disease dataset as a function of learned model size (**a**) and FLOPs (**b**).

Compared with traditional research using lightweight models on plant disease datasets, the compression method introduced in this paper can provide a better tradeoff between the size and accuracy of the model. Moreover, we can manually adjust the pruning rate to obtain the model we want, allowing it to be better deployed in resource-constrained areas.

## 5. Conclusions

In this study, due to the limitations of DNN deployment and the low performance of such networks on resource-constrained devices, the applicability of the resulting models in the field of plant diseases is limited. Therefore, in view of the problems faced by farms in remote areas, the VGGNet and AlexNet models are compressed to reduce the number of required parameters and accelerate reasoning. In the compression method, lightweight fully connected layers, pruning, knowledge distillation, and quantization are combined to obtain efficient models with accuracy losses that fall within an acceptable range. The experimental results also show the effectiveness of using knowledge distillation in iterative pruning. Moreover, the compressed model is more suitable for deployment in resource-constrained areas than the lightweight model.

In the following work, on the one hand, it will also be a key issue to find the optimal pruning number of each convolution layer to achieve a balance between accuracy loss and model size and to achieve the best knowledge distillation effect. On the other hand, extending the dataset as much as possible, compressing the models on field-programmable gate arrays (FPGAs), and deploying them in a real farm environment are additional tasks that need to be carried out.

**Author Contributions:** Methodology, R.W. and W.Z.; software, J.D.; data curation, M.X. and M.W.; writing—review and editing, R.W. and W.Z.; supervision, Y.R. and Z.J. All authors have read and agreed to the published version of the manuscript.

**Funding:** This research was funded by the Key Research and Development Project of Anhui Province (1804a07020108, 201904a06020056, 202104a06020012), Independent Project of Anhui Key Laboratory of Smart Agricultural Technology and Equipment (APKLSATE2019X001), the Ministry of Agriculture Agricultural Internet of Things Technology Integration and Application Key Laboratory Open Fund in 2016 (2016KL05), the Major Science and Technology Special Plan of Anhui Province in 2017 (17030701049) and Major Project of Natural Science Research in Universities of Anhui Province (KJ2019ZD20).

**Institutional Review Board Statement:** Not applicable.

**Informed Consent Statement:** Not applicable.

**Data Availability Statement:** Publicly available datasets were analyzed in this study. This data can be found here: https://www.kaggle.com/abdallahalidev/plantvillage-dataset (accessed date: 2 April 2021).

**Acknowledgments:** We would like to thank Wu Zhang for his help in revising the paper. We are grateful to the reviewers for their suggestions and comments, which significantly improved the quality of this paper.

**Conflicts of Interest:** All the authors declare no conflict of interest.

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
