# Peer review of "Deep Neural Network Compression for Plant Disease Recognition"

_symmetry, doi:10.3390/sym13101769_

Round 1

Reviewer 1 Report

Kindly find my comments in the attached file.

Reviewer 2 Report

This paper presents a study on the compression on deep neural network, proposing a new method to reduce the computation complexity through the use of lightweight fully connected layers, pruning, the use of knowledge distillation, and the compression of the model using quantization. The experiments are conducted in the plant disease detection, using the Plant village dataset.

The topic of this paper could be interesting for this journal and the work is technically solid.

I have the following comments:
- Related Work section should contain also a subsection relative to previous works on plant disease detection. In this new section, I suggest the authors to consider to add the following articles:
       - Deep learning models for plant disease detection and diagnosis. Computers and Electronics in Agriculture
       - A Deep Learning and Social IoT approach for Plants Disease Prediction toward a Sustainable Agriculture. IEEE Internet of Things Journal.
- Why the authors choose only VGGNet and AlexNet? Why they did not consider other architectures such as Xception and InceptionV3 or lightweight architectures such as MobileNetV2? The inclusion of such architectures in the analysis would strengthen the results. 
- Table 4 and 5 could be merged together
- Table 6 and 7 could be merged together
- Table 8 and 9 could be merged together

Finally, a proofread is needed to correct errors and typos. Here, there is a non-exhaustive list:

reduce multiply accumulate -> reduce multiply-accumulate
plant phenotypinge -> plant phenotyping
to current networks with wider, deeper and more parameters -> to current networks with wider, deeper, and more parameters
real farm environment are additional tasks that needs to be carried out -> real farm environment are additional tasks that need to be carried out

Reviewer 3 Report

  1. There is no quantitative comparison with conventional methods (i.e., pruning, quantization, lightweight networks, and knowledge distillation) introduced in Section 2.
  1. What is the difference between the conventional methods and the proposed method? The contribution of this paper is unclear.
  1. Isn't deep neural network compression possible because the data set is easy to classify? It would be nice to apply the proposed method to other data sets that are more difficult to classify.
  1. Authors should add another section on conventional plant disease identification methods. It is necessary to introduce the state-of-the-art methods for plant disease identification. Please refer to the following papers.
  • C. Tetila, Automatic recognition of soybean leaf diseases using UAV images and deep convolutional neural networks, IEEE Geoscience and Remote Sensing Letters 2020, 17, 903-907.
  • J. Yu and C.-H. Son, Leaf spot attention network for apple leaf disease identification, In Proceedings of the IEEE Conference on Computer Vision and Pattern Recognition (IEEE CVPR) Workshops, June 2020.

Round 2

Reviewer 1 Report

Thanks for revising the manuscript according to my suggestions. I would now suggest it for publication in the journal "Symmetry".  

Reviewer 3 Report

Authors have addressed all my comments.